# Long-term cardiology outcomes in children after early treatment for Chagas disease, an observational study

**Nicolás Leonel González** [1]*, **Guillermo Moscatelli**[1,2], **Samanta Moroni**[1],
**Griselda Ballering**[1], **Laura Jurado**[1,2], **Nicolás Falk**[1], **Andrés Bochoeyer**[3],
**Alejandro Goldsman**[3], **María Grippo**[3], **Héctor Freilij**[1], **Facundo Garcia Bournissen**[4],
**Eric Chatelain**[5], **Jaime Altcheh**[1,2]

**1** Servicio de Parasitología y Chagas, Hospital de Niños Ricardo Gutiérrez, Buenos Aires, Argentina,
**2** Instituto Multidisciplinario de Investigación en Patologías Pediátricas (IMIPP) (CONICET-GCBA), Buenos Aires, Argentina, **3** Servicio de Cardiología, Hospital de Niños Ricardo Gutiérrez, Buenos Aires, Argentina,
**4** Division of Pediatric Clinical Pharmacology, Department of Pediatrics, Schulich School of Medicine and Dentistry, University of Western Ontario, London, Ontario, Canada, **5** Drugs for Neglected Diseases initiative (DNDi), Geneva, Switzerland

\* ng211@hotmail.com

**Data Availability Statement:** Data will be available upon approval by the ethics committee of Hospital de Niños Ricardo Gutierrez (email address comite.

## Abstract

### Background

Parasite persistence after acute infection with *Trypanosoma cruzi* is an important factor in the development of Chagas disease (CD) cardiomyopathy. Few studies have investigated the clinical effectiveness of CD treatment through the evaluation of cardiological events by long term follow-up of treated children. Cardiological evaluation in children is challenging since features that would be diagnosed as abnormal in an adult's ECG may be normal, age-related findings in a pediatric ECG trace. The objective was to evaluate cardiac involvement in patients with Chagas disease with a minimum follow-up of 6 years post-treatment.

### Methodology

A descriptive study of a cohort of pediatric patients with CD treated with benznidazole (Bz) or nifurtimox (Nf) was conducted. Children (N = 234) with at least 6 years post CD treatment followed at the Parasitology and Chagas Service, Buenos Aires Children's Hospital (Argentina) were enrolled. By convenience sampling, children who attended a clinical visit between August 2015 and November 2019 were also invited to participate for additional cardiovascular studies like 24-hour Holter monitoring and speckle-tracking 2D echocardiogram (STE). Benznidazole was prescribed in 171 patients and nifurtimox in 63 patients. Baseline parasitemia data was available for 168/234 patients.

During the follow-up period, alterations in routine ECG were observed in 11/234 (4.7%, 95% CI [2–7.4%]) patients. In only four patients, with complete right bundle branch block (cRBBB) and left anterior fascicular block (LAFB), ECG alterations were considered probably related to CD.

bioetica.guti@gmail.com) to investigators who meet the criteria for access to confidential data.

**Funding:** This work was supported by "Fundación para el Estudio de las Infecciones Parasitarias y Enfermedad de Chagas" (FIPEC) Foundation and Drugs for Neglected Diseases initiative (DNDi). DNDi received financial support for this work from UK aid, UK and Fundación Mundo Sano, Argentina; and for its overall mission from Médecins sans Frontières (MSF) and the Swiss Agency for Development and Cooperation (SDC), Switzerland. The funders had no role in study design, data collection and analysis, decision to publish, or preparation of the manuscript.

**Competing interests:** The authors have declared that no competing interests exist.

During follow-up, 129/130 (99%) treated patients achieved persistent negative parasitemia by qPCR. Also decrease in *T.cruzi* antibodies titers was observed in all patients and negative seroconversion occurred in 123/234 (52%) patients.

## Conclusions

A low incidence of cardiological lesions related to CD was observed in patients treated early for pediatric CD. This suggests a protective effect of parasiticidal treatment on the development of cardiological lesions and highlights the importance of early treatment of infected children.

## Trial registration

ClinicalTrials.gov NCT04090489.

## Author summary

If left untreated, CD evolves into a chronic oligosymptomatic infection that can progress to cardiac complications in 30% of patients after several years.

It is known that the main early marker of cardiac involvement are alterations in the conduction system. There are few studies of long-term after treatment cardiological evolution which have assessed the clinical effectiveness of treatment.

The rationale for CD treatment is to avoid the development of cardiological complications. The parasiticidal effect of treatment has been demonstrated but its clinical effectiveness in preventing cardiac involvement requires long term follow-up.

In our long term follow-up study of treated children, we observed the preventive effect on cardiac lesions by treatment with benznidazole or nifurtimox. This intensifies the need for early diagnosis and treatment to prevent the development of long term complications observed in CD.

## Introduction

Chagas disease (CD), caused by *Trypanosoma cruzi* (*T. cruzi*), has a broad range of hosts, and is primarily acquired in childhood through contact with infected triatominae bugs or congenitally. [1] Historically considered a regional disease, CD has now become a global phenomenon due to migration, reaching regions such as Europe, the USA and Japan. [2–5]

CD presents with an acute phase with high parasitemia that, if left untreated, evolves into a chronic oligosymptomatic infection that can progress, in 30% of patients [6], to cardiac and/or gastrointestinal complications after several years. Parasite persistence is important to the development of CD cardiomyopathy. Effective antiparasitic treatment of CD could, therefore, prevent CD cardiac complications. However, there is insufficient data to adequately evaluate the impact of early CD treatment on the prevention of cardiac lesions, since this requires long-term follow-up of treated pediatric patients.

Electrocardiogram (ECG) alterations are early signs of CD cardiac involvement. Some ECG hallmarks of CD are cRBBB, QRS complex widening or fragmentation, and sinus bradycardia. [6,7] New tests like speckle-tracking 2D echocardiogram (STE) has been developed for assess

myocardial contractility deformation. STE is used to detect subtler segmental ventricular wall motion abnormalities an early marker of cardiovascular compromise in CD patients. [8,9]

Few studies have assessed the cardiological outcomes after early treatment of CD in childhood, possibly because cardiological evaluation in children must be conducted by pediatric cardiologist specialists to avoid misdiagnosing age-related findings in pediatric ECG traces as abnormal. Awareness of these differences is the key to correct interpretation of pediatric ECGs.

The aim of this study is to evaluate the incidence of cardiological changes in CD children several years after treatment completion, by ECG, ambulatory 24-hour-ECG monitoring (Holter) as well as the speckle-tracking 2D echocardiogram (STE) to evaluate systolic and diastolic ventricular contractility. Also kinetics of *T. cruzi* antibodies and parasitemia were evaluated as a marker of treatment response.

## Methods

### Ethics statement

The study was approved by the ethics committee of the Hospital de Niños Ricardo Gutiérrez, Buenos Aires, Argentina (CEI N˚16.29). Written informed consent by the patient and/or parents was obtained for all patients according to age and local regulations. The protocol was registered in ClinicalTrials.gov, NCT04090489.

The identity of the patients was kept confidential, as well as the results of the studies carried out by means of a code of numbers and letters.

A descriptive study of cardiological, parasitological and serological results in a cohort of pediatric CD patients, with at least 6 years' post-treatment follow-up, treated with benznidazole (Bz) or nifurtimox (Nf) was conducted at the Servicio de Parasitología y Chagas, Hospital de Niños Ricardo Gutiérrez, Buenos Aires, Argentina. Additionally, by convenience sampling, children who attended a clinical visit between August 2015 and November 2019 were also invited to participate for additional cardiovascular studies (Holter and STE) (Fig 1).

Also, functional gastrointestinal disorders were evaluated using the Rome IV questionnaire. This questionnaire includes alarm symptoms or red flags to alert clinicians to consider additional testing to investigate the presence of megaesophagus or megacolon.

Since the standard of care for CD is the treatment of all infected children, it was not possible to include a non-treated control group due to ethical issues.

Diagnosis criteria for CD: In infants younger than 8 months, CD was diagnosed by direct observation of T. cruzi using a parasitological concentration method (microhematocrit test—MH) or xenodiagnosis, and in infants older than 9 months, by two reactive serological tests (ELISA, and indirect hemagglutination—IHA).

### Inclusion criteria

- Children with CD treated with Bz or Nf with at least 6 years of post-treatment follow-up.

- Signed informed consent or assent according to patient age.

### Exclusion criteria

- Patients with chronic diseases (renal, hepatic, neurological) that could affect the interpretation of the results (at the discretion of the researcher).

- Subjects with congenital heart disease.

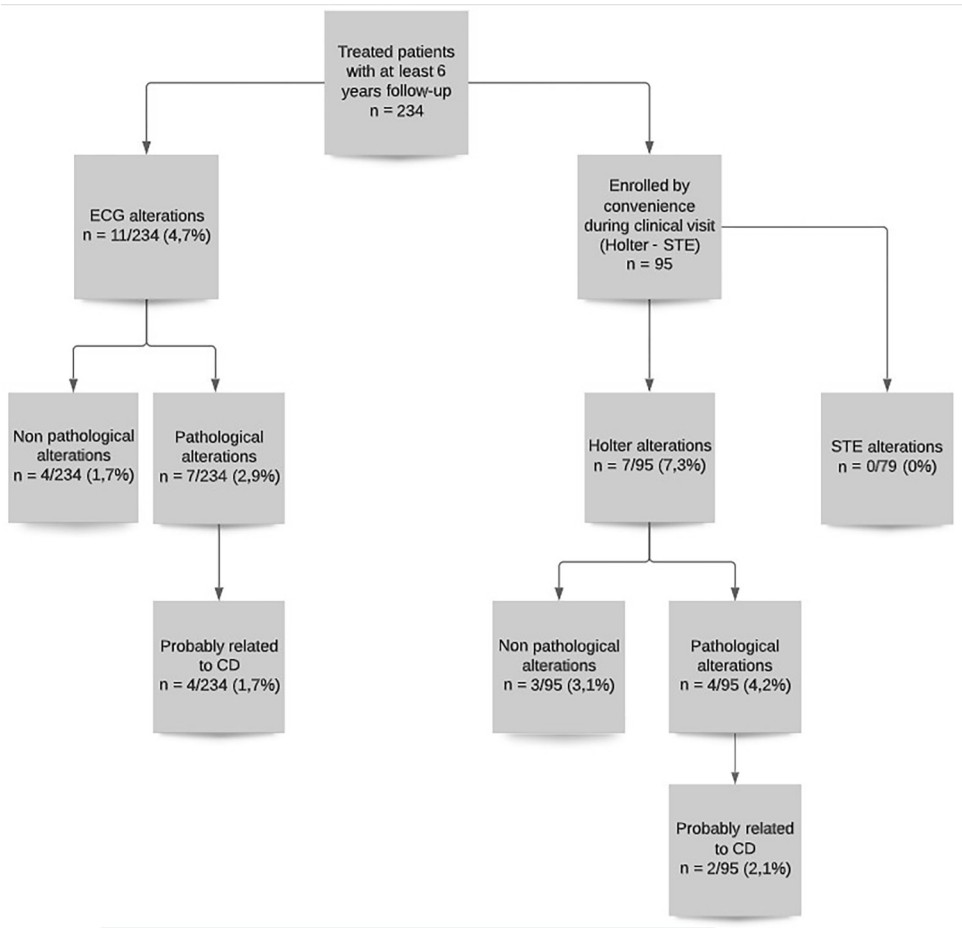

**Fig 1. Flow chart of patient enrollment in the study.**

## Treatment

Bz (5–8 mg/kg in 2 or 3 doses) or Nf (10–12 mg/kg in 2 or 3 doses) was prescribed for 60–90 days at the time of diagnosis. Medication was provided in monthly batches and compliance was assessed by counting the remaining tablets at each visit.

## *T. cruzi* serology and qPCR

Blood samples were collected at diagnosis, end-of-treatment and every 6–12 months thereafter. Detection of *T. cruzi*-satellite DNA region by PCR was carried out in blood (2 mL) mixed with EDTA-guanidine buffer (10). Serological tests were performed by indirect hemagglutination (IHA) (Chagatest HAI; Wiener lab, Rosario, Argentina) and ELISA (Chagatest ELISA lisado; Wiener lab, Rosario, Argentina tests).

## Cardiological evaluation

ECG was performed at diagnosis and every year following treatment, as part of post-treatment follow up protocols. Additionally, by convenience sampling, children who attended a clinical visit between August 2015 and November 2019 were also invited to participate for additional cardiovascular studies (24 hours ECG-Holter and STE).

Evaluation of myocardial strain by STE was performed with a Siemens Acuson SC 2000 echograph and a 2–5MHz cardiac and three-dimensional matrix transducer. The analysis measured cardiac dimensions and cardiac ventricular function using two-dimensional M-mode, Doppler and tissue Doppler imaging with emphasis on measuring longitudinal strain in the left ventricle by speckle tracking. Two-dimensional apical four-chamber view was used to measure global left ventricular peak systolic longitudinal strain and longitudinal systolic strain rate. At least four beats were recorded for later measurements. Encirclement of the ventricle was manually done at the endocardial level. Three-dimensional left ventricle ejection fraction and volume were automatically calculated using Siemens software, in four-chamber views.

Cardiological studies were carried out at the Cardiology service of the Hospital de Niños Ricardo Gutiérrez, by trained pediatric cardiology specialists and pediatric electrophysiologists who were blind to the patients' treatment history.

### Statistical analysis

Continuous variables are presented as means with $CI_{95\%}$ or medians and interquartile range, and categorical variables as percentages.

The kinetics of *T. cruzi* serum antibodies were analyzed using survival analysis. Analyses were performed with R software v3.0 (R Core Team 2018, R Foundation for Statistical Computing, Vienna, Austria https://www.R-project.org/).

### Results

A total of 234 patients (110 males, 124 females) with at least 6 years follow-up were evaluated. The median age at diagnosis was 62.53 months (range 1–202 months, IQR 11–116), with a median follow up time of 9.8 years (6–20.2 years, IQR 7.5–12.4) (Fig 1).

During the follow-up period, alterations in routine ECG were observed in 11/234 (4.7%, 95% CI [2–7.4%]).

However, in 4/11 patients these ECG variations were considered normal variations for children. In the remaining 7/11 patients the ECG variations were considered pathological and only 4/7 patients with cRBBB and LAFB were considered to have ECG abnormalities probably related to CD. Table 1.

In addition, by convenience sampling, a total of 95/234 (40.59%) patients who attended their annual clinical check-up between August 2015 and November 2019 were enrolled for a more in-depth cardiological evaluation with Holter, Doppler echocardiography and STE. 7/11 patients with alterations in routine ECG are found in this group.

Holter monitoring showed pathological findings in 4/95 (4.2%; 95% CI [1.3–11%]) patients. The pathological findings were: isolated ventricular extrasystoles and nocturnal sinus bradycardia (one patient); asymptomatic and vagal related 1st and 2nd degree AV block (one patient); LAFB (one patient); and cRBBB (one patient). Only the last two findings were considered probably related to CD involvement. In these 4 patients, in order to stress myocardial function, ergometer exercise tests with oxygen consumption estimation were conducted, showing normal results.

Another 3/95 (3.1%) patients presented non-pathological findings such as iRBBB).

Myocardial contractility was evaluated by STE in 79/95 (83%) patients (16 patients were not available for evaluation due to missed appointments). STE results: Mean four-chamber global longitudinal strain was -19% while strain rate gave -1 (seg-1). The median longitudinal strain was 19.19%, while for strain rate was -1 (seg-1). Values were found to be above (i.e., less negative) -16% strain in 8 patients. As regards strain rate, 3 patients presented values above

**Table 1. Description of patients with ECG alterations.**

| Age at treatment | Age at evaluation | Treatment | Serology at final evaluation | ECG findings | Other cardiological evaluation | Interpretation |
|---|---|---|---|---|---|---|
| 3 years | 19 years | Bz 70 days Good compliance | ELISA / IHA: Negative | Complete right bundle branch block | STE: Normal | Pathological, probably related to CD. |
| 11 years | 27 years | Bz 60 days Good compliance | ELISA: Reactive 3.1 IHA: 32 | Complete right bundle branch block | - | Pathological, probably related to CD. |
| 11 years | 22 years | Bz 60 days Good compliance | ELISA: Reactive 3.3 IHA: Negative PCR: Negative | Isolated ventricular extrasystoles and nocturnal sinus bradycardia | STE: Normal | Pathological, non related to CD. |
| 5 years | 12 years | Nf 60 days Good compliance | ELISA / IHA: Negative PCR: Negative | 2nd degree AV block | Echocardiogram: Normal | Pathological, non related to CD. |
| 1 years | 11 years | Bz 60 days Good compliance | ELISA / IHA: Negative PCR: Negative | Asymptomatic and vagal related 1st and 2nd degree AV block | STE: Normal | Pathological, non related to CD. |
| 2 years | 13 years | Bz 60 days Good compliance | ELISA / IHA: Negative PCR: Negative | Incomplete right bundle branch block | STE: Normal | Normal pediatric ECG variations. |
| 1 years | 7 years | Bz 60 days Good compliance | ELISA / IHA: Negative PCR: Negative | Incomplete right bundle branch block | STE: Normal | Normal pediatric ECG variations. |
| 10 months | 10 years | Bz 60 days Good compliance | ELISA / IHA: Negative PCR: Negative | Incomplete right bundle branch block | STE: Normal | Normal pediatric ECG variations. |
| 7 years | 13 years | Bz 60 days Good compliance | ELISA: Reactive: 5.9 / IHA: 16 PCR: Negative | Incomplete right bundle branch block | Echocardiogram: Normal | Normal pediatric ECG variations. |
| 6 years | 16 years | Bz 60 days Good compliance | ELISA: Reactive: 1.8 / IHA: Negative PCR: Negative | Left anterior fascicular block | STE: Normal | Pathological, probably related to CD. |
| 10 years | 16 years | Bz 80 days Good compliance | ELISA: Reactive: 11.9 / IHA: 512 PCR: Negative | Left anterior fascicular block | Echocardiogram: Normal | Pathological, probably related to CD. |

-0.8 (seg-1). These cases were considered normal since the global echocardiographic contractility evaluation was normal.

Left ventricular ejection fraction was measured by three-dimensional echocardiography. Mean ejection fraction was 59.2% (45 to 74%).

In conclusion, no disturbances in myocardial contractility were observed by STE and three-dimensional echocardiography. Furthermore, no clinical symptoms such as ventricular tachycardia, syncope, chest pain or palpitations were observed during the follow-up.

## Evaluation of parasiticidal treatment response

In the 234 patients, Bz was prescribed in 171 patients and Nf in 63 patients. Mean Bz dose was 6.6 mg/kg per day (range 4.6–9 mg/kg/day) divided in two (n = 156) or three doses (n = 15). Mean Nf dose was 11.28 mg/kg per day (range 7.2–15 mg/kg/day) divided in two (n = 48) or three doses (n = 15). Only 3 patients did not receive a complete treatment for ChD.

Baseline parasitemia data were available for 168/234 patients: 140/168 (83.3%) were initially positive (91 by qPCR, 7 by both qPCR and MH, 31 by MH and 11 by xenodiagnosis), depending on the availability of parasitological tests. 130/140 positive patients were followed up by parasitemia. In 129/130 (98.5%) parasitemia became negative by T. cruzi—qPCR at the end of treatment and remained persistently negative throughout the follow-up.

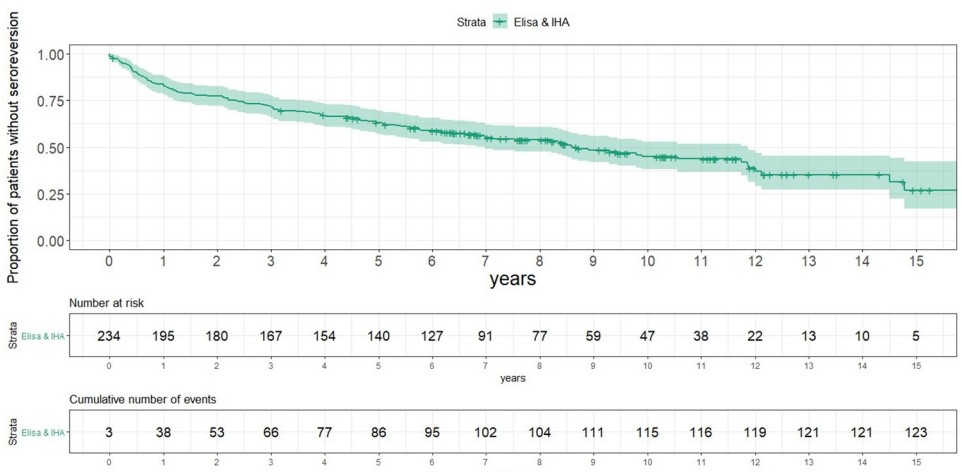

**Fig 2. Serological seroconversion by ELISA and IHA profile in 234 treated patients (Kaplan-Meier).**

A decrease in *T. cruzi* antibody titers by conventional serology (IHA and ELISA) was observed after treatment. Both serological tests became negative in 123 patients (52.56%) at a median time of 8,59 years (95% CI: 6.97–11.86 years). Fig 2.

Regarding functional gastrointestinal disorders, Rome IV diagnostic questionnaire did not show any relevant gastrointestinal symptoms that would require additional gastrointestinal evaluation in any patients.

## Discussion

There are few studies that have investigated the clinical effectiveness of CD treatment, by means of the evaluation of cardiological events, in a long-term follow-up of asymptomatic treated children.

To our knowledge this is the first study of a cohort of treated patients where parasitological, serological and cardiological results, using the latest cardiological available techniques, were evaluated in conjunction with one another in a long term follow-up. A decrease in *T. cruzi* antibodies measured by conventional serology and persistent non-detection of *T. cruzi* by qPCR during after-treatment follow-up was observed in CD treated patients, with an average post-treatment follow-up of 10 years. These results showed the parasiticidal effect of treatment and are in line with previous studies published by our group [10,11], and also studies of other groups [12,13].

No related CD changes in cardiac function, evaluated by ECG and 24-hour ambulatory ECG monitoring and STE, were found in the majority of treated patients. It is likely that this was related to the protective effect of treatment at an early stage of *T. cruzi* infection.

Currently, CD treatment with Bz or Nf is recommended as the standard of care for acute and early chronic phase in children. In the early phase of infection, which is more easily observed in children, the majority of cases are asymptomatic and without cardiac involvement, as described in this study.

Multiple mechanisms may be responsible for the development of cardiac lesions in *T. cruzi* infection. The presence and persistence of the parasite is a critical factor which triggers a specific immune response, inducing vascular endothelial cell damage. [14] The treatment of infected subjects is, therefore, based on the elimination of intracellular parasites to avoid the development of future cardiological complications. [15–17] Several studies have shown that in

non-treated adult CD patients [18–23], the rate of cardiac disease progression was between 13 and 25% with a 1.48 to 4% annual progression rate.

There are some controversial results about the clinical effect of treatment in adults. The most relevant question about CD treatment is whether it will have a clinical benefit preventing the development of cardiological lesions. A pivotal study by Viotti [18] described the beneficial treatment effect in adult patients with chronic CD, changing the perspective about the clinical effect of CD treatment. Others studies, mainly retrospective, showed positive effects of treatment in the progression of heart disease. [13,19,20,24–26]

Recently a retrospective cohort study reinforced previous findings showing that treated patients had fewer ECG alterations (7.9% vs 21.1%) compared with the non-treated group. [27].

However, in the BENEFIT study [28] no difference between treated subjects and the placebo group was observed in adult patients with advanced chronic CD related cardiac complications. This study showed that in advanced chronic CD the cardiological damage was probably irreversible and not modifiable by the parasiticidal treatment.

Regarding children, the available information is based on outdated individual studies and reports vary substantially due to diverse design, small sample size and quality of the studies. However, some positive results about treatment benefits have been published. [12,13,29]

As regards the evaluation of cardiac compromise by ECG it is essential to take into account that children are different from adults. Conduction intervals (PR interval, QRS duration) are shorter than adults due to smaller cardiac size. First degree AV block and Wenckebach sequence may be a normal finding in healthy children during rest or under vagal tone. The QT interval depends on heart rate and age. Heart rates are much faster in neonates and infants, decreasing as the child grows older. Thus, cardiological evaluation in children should be conducted by pediatric cardiologists. In addition, some pathological findings could be related to technical issues and in other cases may be related to congenital lesions and not produced by *T. cruzi* infection.

Awareness of these differences is the key factor for the correct interpretation of pediatric ECG. [30] For example, iRBBB can be a normal ECG alteration during infancy and early childhood. [31] A prevalence of 0.75% for cardiac conduction disturbance including 0.32% prevalence of iRBBB and 0.11% cRBBB was reported in children without any clinical significance. [32] A controversial long-term follow-up study of treated children with Bz for 60 days was not associated with fewer ECG abnormalities as compared to no treatment. Also, the prevalence of ECGs with abnormalities was higher among treated children compared with those not treated [33]. However, it should be emphasized that most of the ECG alterations described were normal findings in children (i.e. extrasystoles, iRBBB, first-degree atrioventricular block) which are unlikely to be related to CD. Only 4/94 children showed alterations of right bundle conduction that could be attributed to CD. This is a clear example of inadequate evaluation of ECG in children, as was seen in several published studies reporting ECG abnormalities in children, which are unlikely to be related to CD. [29,34,35]

Recently a retrospective evaluation of treated children showed differences in ECG alterations between treated and non-treated (4/41 (9.75%) vs 9/41 (21.95%)) patients [36]. However, in contrast to our study, only very few patients were evaluated by 2D echocardiogram or Holter.

In our cohort, with a longer after-treatment follow-up, only four patients showed ECG alterations that could possibly be related to *T. cruzi* infection (cRBBB, LAFB). Regarding the patients with LAFB, it can have many possible causes, and it is not as pathognomonic alterations of CHD. There are other causes that could justify the appearance of these ECG alterations. However, the functional echocardiographic studies were normal. The other ECG

findings observed were in fact likely to be normal ECG findings in children or therefore unrelated to CD. In order to confirm these results, the patients will continue in a prospective follow-up.

New techniques, like STE, have been developed to measure the regional strain percentage of deformation between two ventricle sites during contraction, and strain rate is the speed at which this deformation occurs. In our study using strain and strain rate measurement techniques, treated patients did not show significant long-term regional wall motion. To our knowledge, there were no previous reports of these measurements by STE in pediatric CD treated patients.

Functional gastrointestinal disorders were described in approximately 10% of patients after several years of infection. In order to evaluate gastrointestinal dysfunction, the Rome IV diagnostic questionnaire was used. No relevant gastrointestinal symptoms that required further gastrointestinal evaluation in any patient were found.

A weakness of our study is that it lacks an untreated control group due to the fact that the standard of care for pediatric CD is the treatment of infected children as soon as possible after diagnosis. Also not all treated patients were available for by Holter and STE evaluation. However, our study shows the best available evidence that cardiomyopathy progression rate was lower than in previously reported studies [23,27], reinforcing the preventive treatment effect of CD.

As was previously reported by our group [10,37,38], treated patients showed parasitological and serological signs of treatment response (a decrease in antibody titers, seroreversion in more than half of the patients and constant negative T. cruzi—qPCR). More importantly, we have shown, using the most sensitive cardiological techniques available, that the development of cardiac alterations was most probably prevented by parasiticidal treatment after a median follow-up of over a decade. These results further underline the importance of early diagnosis and availability of treatment, especially for children, young adults and women of childbearing age, in order to prevent new infections and the development of symptomatic cardiac chronic CD.

Further studies with long-term follow-up (20–30 years) of the present cohort will be conducted in order to confirm that early treatment prevents later cardiological compromise in CD patients.

## Supporting information

**S1 File. STROBE Statement.** Checklist of items that should be included in reports of cohort studies.
(DOC)

## Author Contributions

**Conceptualization:** Nicolás Leonel González, Jaime Altcheh.

**Formal analysis:** Nicolás Falk, Facundo Garcia Bournissen.

**Investigation:** Nicolás Leonel González, Griselda Ballering, Laura Jurado.

**Methodology:** Nicolás Leonel González.

**Resources:** Guillermo Moscatelli, Samanta Moroni, Andrés Bochoeyer, Alejandro Goldsman, María Grippo.

**Supervision:** Nicolás Leonel González, Héctor Freilij, Jaime Altcheh.

**Visualization:** Nicolás Leonel González, Eric Chatelain, Jaime Altcheh.

**Writing – original draft:** Nicolás Leonel González, Jaime Altcheh.

**Writing – review & editing:** Nicolás Leonel González, Guillermo Moscatelli, Samanta Moroni, Griselda Ballering, Laura Jurado, Nicolás Falk, Andrés Bochoeyer, Alejandro Goldsman, María Grippo, Héctor Freilij, Facundo Garcia Bournissen, Eric Chatelain, Jaime Altcheh.

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
