## [Decision Letter · Decision Letter 0]

7 Oct 2022

Dear Dr González,

Thank you very much for submitting your manuscript "Long-term cardiology outcomes in children after early treatment for Chagas disease" for consideration at PLOS Neglected Tropical Diseases. As with all papers reviewed by the journal, your manuscript was reviewed by members of the editorial board and by several independent reviewers. The reviewers appreciated the attention to an important topic. Based on the reviews, we are likely to accept this manuscript for publication, providing that you modify the manuscript according to the review recommendations. 

In addition of addressing reviewers' comments, we ask you also to consider that this is an observational study and the STROBE statement and checklist should be followed instead of the TREND statement.

Sincerely,

Andrea Angheben

Academic Editor

Ricardo Fujiwara

Section Editor

Reviewer's Responses to Questions

**Key Review Criteria Required for Acceptance?**

**Methods**

-Are the objectives of the study clearly articulated with a clear testable hypothesis stated?

-Is the study design appropriate to address the stated objectives?

-Is the population clearly described and appropriate for the hypothesis being tested?

-Is the sample size sufficient to ensure adequate power to address the hypothesis being tested?

-Were correct statistical analysis used to support conclusions?

-Are there concerns about ethical or regulatory requirements being met?

Reviewer #1: The objectives of the study are clearly stated. They use cardiological, parasitological and serological assessments to analyze the final outcomes of the cohort of children diagnosed with CD. 

The study design is the most appropriate to address the objectives. The population is clearly described. 

They also describe the median time of follow-up per patient. However, I missed the total follow-up time of the whole cohort. Additonally, when they are talking about diagnosis criteria for CD they do not include PCR as an option, mainly when later on the text (line 141) the serology and PCR are included to assess the parasitological results. If the endpoint is to determine the negativity of the PCR test for T cruzi, why do not include this test as diagnosis criteria for CD?

Regarding the analysis, I was wondering if the authors could add a table to compare outcomes between younger children (<12m-old) and the rest of the cohort for the serological, parasitological and cardiological results. I'm asking for this because the authors state that an early diagnosis is crucial for the final outcome regarding cardiological involvement. 

Ethics statement do not show a sentence regarding data protection regulation issues.

Reviewer #2: The results of this article provide extremely interesting data on a subgroup of patients on which there is not much published work as is the case of children. the article is well written and reflects the effort and work of a group of researchers of consolidated name. 

The children were treated and clinically accompanied with a wide variety of serological techniques, molecular biology and complementary tests, which is undoubtedly of the highest scientific value.

my only comment I am going to make about the article is that patients treated and accompanied for an average of 10 years are found practically no cardiological event. the authors attribute the absence of these events to the effect of the drug but in my opinion, there is no basis to think that it was the drug itself since there is no control group or comparator. the authors explain well that it would not be ethically accepted to have patients diagnosed with Chagas disease and not offer them treatment, but it would have been interesting to have had a cohort of patients without Chagas disease who had the same follow-up protocol. 

On the other hand, it would be very clarifying to be able to understand based on which criteria the electrocardiographic alterations have been attributed or not to Chagas disease.

Reviewer #3: The study design is appropriate and the objectives clearly articulated

**Results**

-Does the analysis presented match the analysis plan?

-Are the results clearly and completely presented?

-Are the figures (Tables, Images) of sufficient quality for clarity?

Reviewer #1: The results are clearly presented, with a good reading of all of them. As I said in the methods, I missed a table to show the results comaring younger children and the rest of the cohort, including a p-value (likely to be not statistically significant) in it. 

Median follow-up time per patient is 9.8y, but what about the total follow-up time of the cohort? I would include this result in the first paragraph of this section. 

The median age at diagnosis of CD is 62.5m with a range of 1 to 202 months (which IQR has the age at diagnosis?). This data is surprising for me because I had understood from previous CD studies that the best age for starting the treatment of CD should be when children are <12m-old. Have the authors observed any differences between thes age-groups?

I was wondering about the 3 cases (table 1) with abnormal ECG findings that were not considered related to CD, why do you not consider these findings related to CD? Case 1 was diagnosed at 3y and the evaluation was performed at 19y. This case serorreverted (ELISA result is neg) but you considered that cardiological findings were related to CD, on the contrary other cases with positive serological results had cardiological findings that were not considered related to CD, could you better explain these issues to the readers? Thank you. 

Any age-related outcomes? I would add a table showing the differences.

Has the age at diagnosis an impact for the final parasitological or serological or cardiological outcomes?

In line 221 the same sentence is repeated twice.

Reviewer #2: see above

Reviewer #3: Overall they found ECG abnormalities in 4.7% of subjects (11/234) but according to their interpretation only 2/234 (0.8%) were related to Chagas disease. Although I’m not a cardiologist I’m not sure that “Left anterior fascicular block (LAFB)” observed in two subjects (both 16 years-old) can be regarded as “normal pediatric ECG abnormalities” and not as CD- related alterations. LAFB is commonly considered an early ECG sign of Chagas cardiomyopathy, both subjects still have an ELISA reactive test and probably as for their age should be considered adults. Are the authors able to produce more convincing evidence that such patients should be dismissed as not having CD-alterations? 

In table 1 could be useful add a column reporting other cardiological evaluation (i.e. Holter, echo, STE) if such exams were done.

Figure 1 is unclear how many of the 11 patients with ECG alterations underwent Holter or STE.

Regarding treatment can you report how many patients completed the scheduled therapy in both groups?

Page 15 line 221-222 the sentence is reported twice.

Page 15 line 224-225: “A decrease in T. cruzi antibody titers by conventional serology (IHA and ELISA) was observed after treatment. This is not reported and I think you should provide such data.

**Conclusions**

-Are the conclusions supported by the data presented?

-Are the limitations of analysis clearly described?

-Do the authors discuss how these data can be helpful to advance our understanding of the topic under study?

-Is public health relevance addressed?

Reviewer #1: The conclusions are supported by the data presented. 

However, in line 244 the authors comment on early treatment but the median age at diagnosis for the cohort was not as early as recommended by international health institutions such as the WHO. 

The supporting information sheet is empty and I cannot evaluate it. 

The public health relevance is clearly addressed by the authors.

Reviewer #2: see above

Reviewer #3: Yes the conclusions are supported by the data presented but the number of patients with cardiological alterations should be changed (see my comments on the results)

Only 41% subjects underwent Holter and STE; this should be reported among the study limits

**Editorial and Data Presentation Modifications?**

Reviewer #1: (No Response)

Reviewer #2: (No Response)

Reviewer #3: Some references are outdated or inappropriate (for instance ref. 2,3,4,5). Please consider the following:

Bern C et al. Chagas disease in the United States: a public health approach. Clin Microbiol Rev 2019;23(1):e00023-19.

Abras A et al. Worldwide control and management of Chagas disease in a new era of globalization: a close look at congenital Trypanosoma cruzi infection. Clin Microbiol Rev 2022; 35(2):e0015221.

Imai K, et al. Chagas disease : a report of 17 suspected cases in Japan, 2012-2017. Trop Med Health 2019;47:138.

Freitas Lindani KC et al. Chagas disease from discovery to a world wide health problem. Front Public Health 2019;7:166.

**Summary and General Comments**

Reviewer #1: (No Response)

Reviewer #2: In summary, paper with a very high scientific value on a population of which we are not used to having data but the authors would have to minimize or at least contextualize the effect of the drug on the clinical evolution of patients

Reviewer #3: This is an interesting study conducted in Buenos Aires (Argentina) in a cohort of children aiming to evaluate the cardiological outcome post-treatment for Chagas disease with a median follow-up of 9.8 years. 

As reported by the authors this issue has been previously evaluated in a limited number of studies some of which biased by the lack of appropriate ECG interpretation by a pediatric cardiologist.

Only 41% subjects underwent Holter and STE; this should be reported among the study limits

PLOS authors have the option to publish the peer review history of their article (what does this mean?). If published, this will include your full peer review and any attached files.

Reviewer #1: Yes: Antoni Soriano-Arandes

Reviewer #2: No

Reviewer #3: No

Figure Files:

Data Requirements:

Reproducibility:

References

---

## [Editor Report · Decision Letter 1]

21 Nov 2022

Dear Dr González,

We are pleased to inform you that your manuscript 'Long-term cardiology outcomes in children after early treatment for Chagas disease' has been provisionally accepted for publication in PLOS Neglected Tropical Diseases.

In order to proceed with the process of publication, we ask that you make two minor typing corrections: at line 25 Trypanosoma cruzi should be in italics, and at line 31 please erase the " ' " after years. Moreover, to comply with the STROBE requirements, please adjust the title to "Long-term cardiology outcomes in children after early treatment for Chagas disease, an observational study".

Before your manuscript can be formally accepted you will also need to complete some formatting changes, which you will receive in a follow up email. A member of our team will be in touch with a set of requests.

Best regards,

Andrea Angheben

Academic Editor

Ricardo Fujiwara

Section Editor

---

## [Editor Report · Acceptance letter]

13 Dec 2022

Dear Dr González,

We are delighted to inform you that your manuscript, "Long-term cardiology outcomes in children after early treatment for Chagas disease, an observational study," has been formally accepted for publication in PLOS Neglected Tropical Diseases.

Best regards,

Shaden Kamhawi

co-Editor-in-Chief

Paul Brindley

co-Editor-in-Chief
